# Mitochondria: A Galaxy in the Hematopoietic and Leukemic Stem Cell Universe

**DOI:** 10.3390/ijms21113928

**Published:** 2020-05-30

**Authors:** Cristina Panuzzo, Aleksandar Jovanovski, Barbara Pergolizzi, Lucrezia Pironi, Serena Stanga, Carmen Fava, Daniela Cilloni

**Affiliations:** 1Department of Clinical and Biological Sciences, University of Turin, 10043 Orbassano, Italy; aleksandar.jovanovski@unito.it (A.J.); barbara.pergolizzi@unito.it (B.P.); lucrezia.pironi@edu.unito.it (L.P.); carmen.fava@unito.it (C.F.); 2Department of Neuroscience Rita Levi Montalcini, 10124 Turin, Italy; serena.stanga@unito.it; 3Neuroscience Institute Cavalieri Ottolenghi, University of Turin, 10043 Orbassano, Italy

**Keywords:** mitochondria, mitochondrial dysfunction, oxidative phosphorylation program, reactive oxygen species (ROS), apoptosis, leukemia, hematopoietic stem cell, leukemic stem cell, mitophagy, mitochondrial targeted therapy

## Abstract

Mitochondria are the main fascinating energetic source into the cells. Their number, shape, and dynamism are controlled by the cell’s type and current behavior. The perturbation of the mitochondrial inward system via stress response and/or oncogenic insults could activate several trafficking molecular mechanisms with the intention to solve the problem. In this review, we aimed to clarify the crucial pathways in the mitochondrial system, dissecting the different metabolic defects, with a special emphasis on hematological malignancies. We investigated the pivotal role of mitochondria in the maintenance of hematopoietic stem cells (HSCs) and their main alterations that could induce malignant transformation, culminating in the generation of leukemic stem cells (LSCs). In addition, we presented an overview of LSCs mitochondrial dysregulated mechanisms in terms of (1) increasing in oxidative phosphorylation program (OXPHOS), as a crucial process for survival and self-renewal of LSCs,(2) low levels of reactive oxygen species (ROS), and (3) aberrant expression of B-cell lymphoma 2 (Bcl-2) with sustained mitophagy. Furthermore, these peculiarities may represent attractive new “hot spots” for mitochondrial-targeted therapy. Finally, we remark the potential of the LCS metabolic effectors to be exploited as novel therapeutic targets.

## 1. Introduction

Defined in biological vocabulary, mitochondria are “Perpetuum mobile” of the cell and its source of energy. However, speaking philosophically, mitochondria are “cells” in the cell itself, a compartmentalized “small organisms” with their own deoxyribonucleic acid (DNA). As specified by endosymbiotic theory [1], they originate from prokaryotic cells—alpha-proteobacteria, which during the evolution were imbibed by eukaryotic cells [2]. The truth of this theory is sustained by the fact that mitochondria contain their own genetic material, a circular genome that represents mitochondrial DNA (mtDNA), also encoding proteins [3,4,5]. During their maximal activity, they are elongated and interconnected organelles; consequently, the shape of the mitochondria reveals the degree of cell energetic activity [6,7,8]. Although they have many aspects of autonomy, their biogenesis depends on the inevitable cooperation with other organelles, especially with the endoplasmic reticulum (ER). This structural-physiological unit of communication between mitochondria and the ER is named mitochondria-associated membranes (MAM). Through this communication fissure with ER, mitochondria import essential resources that serve as dynamic sustainability of their structures, such as lipid precursors like phosphatidylserine, phosphatidic acid, phosphatidylcholine, phosphatidylinositol, and sphingolipids [9]. Finally, alongside the unique structural and metabolic characteristics, mitochondria are the major cause of reactive oxygen species (ROS) production in response to cellular stress and this event can induce cellular damage. Thus, a quality check process is crucial to prevent the onset of several pathological conditions. Within this scenario, the interplay between mitochondrial biogenesis and mitochondrial degradation through mitophagy is crucial for mitochondrial homeostasis.

In this comprehensive review, we aimed through an attentive examination of the processes along with the axis normal function—dysfunction to highlight which are the most considerable molecular mechanisms responsible for mitochondrial dysfunction, with particular regard to hematopoietic and leukemic stem cells. In addition, we discuss potential novel treatment alternatives arising from the latest knowledge of impaired metabolic and apoptotic pathways in different leukemic disorders.

### Mitochondria: From Structure to Metabolic Pathways

Mitochondria are constituted of two membranes, which differ in terms of protein composition and function. The outer mitochondrial membrane (OMM) separates mitochondria from the cytosol, and the communication is maintained through porous structures, for instance, the voltage-dependent anion channel (VDAC), which allows metabolites and other substances to pass in both directions [10]. Between the membranes, there is the intermembrane space (IMS), and the area that is bounded by the inner mitochondrial membrane (IMM) is called the mitochondrial matrix. The IMM is the place where the protein complexes of the respiratory chain (RC) are located. They can be seen as both individual orsupercomplexes [11]. IMM has an intensely folded structure, thereby enlarging the surface several times and creating the well-known cristae, one of the main features of mitochondria as organelles [12]. In fact, cristae are much more than just a physical phenomenon in the mitochondrial picture: they undergo an intensive remodeling process, changing the depth of their folds depending on cell activity. During the pro-apoptotic process, mediated by the pro-apoptotic BCL-2 family proteins, cristae undergo intensive remodeling in order to increase the cytochrome c infusion [13]. The shape of the cristae along with the IMM also depends on the energetic status of the cell. During intensive energy production, their twisting can multiply in order to provide more efficient electronic transport [14]. 

Along the membrane, the electron transport over RC and protein pumping into the IMS generates the proton gradient-potential gradient, thus achieving a process of oxidative phosphorylation (OXPHOS), the final stage of cellular respiration [11,12,15]. The electrons that enter the RC originate from the Krebs cycle, also known as the tricarboxylic acid (TCA) cycle. The TCA cycle—eight series of enzymatic reactions—starts with the reaction between acetyl-CoA and oxaloacetate forming citrate. Reduced cofactors, nicotinamide adenine dinucleotide (NADH), and flavin adenine dinucleotide (FADH2) are released during these subsequent reactions. In the final reaction of the cycle, the malate is converted to oxalacetate by transferring two electrons to NADH, and the oxalacetate is ready to re-enter the cycle. The electrons that are generated in the TCA cycle are carried by NADH and FADH2 and then passed to the RC [15]—then, by OXPHOS of these NADH and FADH2 factors, even if with lower efficiency if compared to Krebs cycle.

In addition to the classical pathway responsible for ATP synthesis, another important source of acetyl-CoA is the β-oxidation of fatty acids. Although long-chain fatty acids are oxidized in peroxisomes, short-chain fatty acids can simply diffuse through the IMM, where they are oxidized. Free fatty acids are stored as neutral triacylglycerols (TAGs) in lipid droplets (LD) and can be taken up into mitochondria for energy reserve. The process initiates with the action of the adipose triacylglycerol lipase (ATGL) and hormone-sensitive lipase by which fatty acids are released from TAG. After a series of catabolic processes, the fatty acids and degraded to acetyl-CoA, by providing substrate TCA and EC, even if with lower efficiency if compared to the Krebs cycle.

Besides energy production, mithochondria also have vital roles in numerous cellular processes, including apoptotic activation and cell death, maintenance of ion homeostasis especially calcium, synthesis of phospholipids (for example phosphatidylethanolamine, phosphatidylglycerol, and cardiolipin), of amino acids and heme, among others [16,17].

The mitochondria demonstrate a high demand for lipid precursors since they have two organelle membranes and are involved in the final refinement of the membranes’ lipids. The communication with ER through MAM is a two-way trade and the amplest phospholipids of the cells, phosphatidylcholine, phosphatidylethanolamine, and phosphatidylserine cannot be synthesized without well defined, two-way trading. For example, the phosphatidylserine is first produced by the enzymatic reaction in the ER and then transferred to the IMM to be converted into phosphatidylethanolamine. Otherwise, phosphatidylethanolamine, in order to be modified into phosphatidylcholine, must be relocated to the ER, where the enzymes for this reaction take place.

Cardiolipin (CL) is contemplated as the most important mitochondria-specific phospholipid due to its role in mitochondrial morphology, cellular signalling, RC and metabolism as in a mitochondrial life cycle. The IMM is the place where the biosynthesis of CL is performed. Further to this, many studies have investigated the tissue-specific profile of the CL as well as the factors that could influence these differences [18]. Because of its unique structure and localization, CL has an important role in defining the shape and folding of the IMM, the function of RC and membrane potential gradient as well as the modulation of fission and fusion pathway [19,20].

Finally, mitochondria modulate amino acid homeostasis according to the cellular needs and to the changes in nutrient availability, in order to survive [21]. Thus, mitochondrial metabolism can be modulated even from amino acid availability. Unlike bacteria and plants, mammals have the peculiarity to synthesize in the cytosol, in an ATP-requiring reaction, 11 of 20 aminoacids, named essential aminoacids. Among them, we remembered glutamate, glutamine, proline, and arginine that are directly formed from the Krebs cycle intermediate product α-ketoglutarate. Aspartate and asparaginesreadily formed by mitochondria provided oxaloacetate, while glycine, cysteine, and serineare made from intermediates formed by glycolysis [21]. Besides amino acid synthesis, mitochondria have a central role in amino acid metabolism in order to produce precursors and substrates (carbon backbone of all amino acids) for TCA cycle intermediates, which culminate in the generation of ATP [22]. Indeed, dysfunction of mitochondrial enzymes involved in the metabolic pathways of amino acids is a driver factor for over 40 known disorders in humans. 

## 2. A General Overview of Respiratory Chain Dysfunctions and Apoptotic Dysregulations

Mitochondrial diseases are usually a various group of hereditary aging-related disorders with diverse clinical manifestations on multiple organ systems. In general, the etiology of mitochondrial diseases can be summarized as malfunctioning of the respiratory chain and dysregulation of cell growth, proliferation, and cell survival due to the presence of dysfunctional proteins encoded by mitochondrial or nuclear genetic material [23]. The list of diseases in which mitochondrial dysfunctions represent the major underlying cause or make a significant contribution is broad, and it includes several neurodegenerative and cognitive diseases, certain myopathies, endocrine diseases, such as diabetes mellitus, cardiovascular and renal diseases, as well as many cancer types, including the hematological malignancies [24]. The complex link between leukemogenesis and mitochondrial dysfunction can also be seen in the use of certain drugs, such as the antibiotic chloramphenicol. Indeed, prolonged treatment with chloramphenicol, a known inhibitor of bacterial and mitochondrial protein synthesis, induces dysplastic changes in the bone marrow, sideroblastic anemia and aplastic anemia, which are typical features observed in myelodysplastic syndromes (MDS) and in acute myeloid leukemia (AML) [25]. Indeed, in terms of energy and mitochondria activity, erythropoiesis is one of the most demanding processes in the body. Thus, it is not surprising that anemia is the first effect to be observed when mitochondria are dysfunctional. Furthermore, chloramphenicol causes inhibition of apoptosis, mitochondrial translation, ATP synthesis, respiratory activity, and cell growth. These events increase mitochondrial stress resulting in altered cellular metabolism and activation of further processes, which lead to the onset of leukemia [25,26,27].

### 2.1. Disease-Associated Defects in Oxidative Phosphorylation

The electron transport chain (ETC) is a term that describes the complex processes of simultaneous electronic transfer and redox reactions across four protein complexes (I–IV) embedded within the IMM. Mutations in genes that encode proteins and enzymes of the ETC lead to an irregular process of electron transport, inefficient oxidative phosphorylation, and energy cell deficit [28,29].

Recently, 200 AML patients from the Cancer Genome Atlas (TCGA) dataset were analyzed, showing the presence of 8% of patients with mutations in one or more of ETC complexes genes. It was found that the majority of these mutations affected Complex IV; moreover, they occur more frequently in older patients and they are associated with *TP53* mutations and worse survival rates [30].

Complex I or NADH-coenzyme Q oxidoreductase is a large protein complex built of 46 subunits [31], and NADH dehydrogenase (ND) subunits 1 to 7 (ND1–ND7) are mitochondrially encoded. The implications of Complex I in human pathology are investigated through its ability, together with Complex III, to produce ROS that are toxic to lipids, proteins, and nucleic acids [32]. Some of these ROS are produced during normal cellular metabolism and they have an endogenous origin, mainly by Complex I and III, but there are ROS engendered due to the influence of external factors such as radiation, UV radiation, and heavy metals [33,34]. Although normal ROS production levels play an important role in maintaining proper intracellular signaling, increased levels of ROS production or dysfunction of the cell’s antioxidant defense mechanisms may be the initial trigger factor for a wide range of diseases [35,36]. Functional abnormalities in complex I, which are a consequence of gene mutations (nuclear or mtDNA), lead to increased ROS production and a decrease of the energetic cellular capacity [32,37].

Following this line of events, many studies have shown a link between dysfunctional mitochondrial metabolism, oxidative stress, and some chronic degenerative diseases [38,39,40], for example, the Leber hereditary optic neuropathy [41]. Recent studies, sublimated in one significant meta-analysis, indicate a strong correlation between the etiopathogenesis of Parkinson’s and Alzheimer’s disease and complex I and/or complex IV dysfunction [42,43]. A certain association between mitochondrial impairment and major psychiatric disorders has also been confirmed [43]. *ND* genes code the ND subunits of Complex I, and they are one of the most common mutations related to the Complex I functional impairment. ND mutations were also described in AML patients correlating with shorter overall survival [29,44].

Complex II or succinate dehydrogenase (SDH), besides being part of the respiratory transport chain, contemporaneous is a part of the citric acid cycle. It is built of four subunits (SDHA-D), and unlike other mitochondrial complexes, all four of the subunits are nuclear genes encoded [34,45]. Due to its role in apoptosis and certain types of tumorigenesis [46,47], its pathological implications are related to some neurological diseases such as Leigh syndrome and Huntington’s disease [48,49]. Recently, cysteine depletion has proven effective to target AML cells through a significant impairment of glutathione synthesis, leading to a reduction in glutathionylation of SDHA that, in turn, affects the ETC II activity [32,50].

The next junction in the respiratory chain is complex III or coenzyme Q–cytochrome c reductase. Complex III is an oxidoreductase enzyme that reduces ubiquinone to ubiquinol, which is why the reactions in complex III are also called the ubiquinone cycle [51]. The clinical implications of complex III are closely related to the production of ROS in association with complex I which conjointly contributes to the explanation of the free radicals’ theory of aging [52]. Moreover, a recently published immunological study proved the essential role of mitochondrial complex III in the suppression of regulatory T cells [53]. There is quite a bit of scientific data on the involvement of complex III in human pathology, but we will mention only the most frequent ones. Among these disorders are GRACILE (growth retardation, aminoaciduria, cholestasis, iron overload, lactic acidosis) and Björnstad syndrome; both of them are prompted by a mutation in the mitochondrial chaperon gene *BCS1L* [45,54]. No significant association between Complex III mutations and overall survival in leukemia was found so far.

The enzyme cytochrome c oxidase (COX), or respiratory complex IV, is a membrane protein with a complex structure. Constructed of 14 subunits, it is the final link of the respiratory chain that works closely with cytochrome c [55]. A number of external factors, such as cyanide and CO, affects the activity of complex IV, thereby reducing or completely blocking its participation in electron transfer. This disables the process of oxidative phosphorylation which actually means energy deficiency and cell death [56].

COX deficiency can exist as complete dysfunction of the whole complex or its individual parts, depending on whether the mutations affect the mitochondrial or nuclear genome. Some of the clinical manifestations associated with COX deficiency have been described previously, together with complex I, and they are neonatal tubulopathy, Leigh syndrome, neonatal-onset hepatic failure and encephalopathy, early-onset cardiomyopathy, Alpers–Huttenlocher like disease, ataxia, and muscle hypotonia [57,58].

From the hematological point of view, in MDS pathology, the reduction in expression of mitochondrial genes, and mutations in subunits 1 and 2 of cytochrome c-oxidase (*COI* and *COII*) are well established. They seem to represent the main cause of respiratory chain defect, leading to a reduction in conversion from Fe^3+^ to Fe^2+^ and consequently to sideroblastic anemia [59]. Moreover, the 16% of AML patients with non-synonymous *COII* mutations exhibit worse prognosis and shorter disease-free survival (DFS), especially in normal cytogenetics AML (CN-AML) patients. On the whole, these experimental findings might be exploited as novel prognostic markers to be explored in this subset of AML patients [60]. The co-occurrence of mutations in ETC complex I and IV further increase the adverse prognosis, with a significant reduction in overall survival [30]. 

Interestingly, D-2-hydroxyglutarate (2-HG) accumulation, a typical feature in the isocitrate dehydrogenase (*IDH)* AML mutated cells, negatively impact complex IV with a significant reduction of COX enzymatic activity, when compared to *IDH* wild type AML cells. Moreover, combinatory therapy targeting mitochondrial oxidative phosphorylation significantly improves the efficacy of IDH mutant inhibitors in AML patients [61]. Mutations in *IDH* genes result in alteration of cellular metabolism, impaired cell adaptation mechanisms, epigenetic dysregulation, and they are associated with a poor prognosis in patients with AML [62,63,64,65]. The IDH enzymes exist in three homodimeric isoforms, IDH1, IDH2, and IDH3. Both IDH1 and IDH2 play an enzymatic role in the TCA cycle, converting isocitrate to α-ketoglutarate (α-KG). Reduced nicotinamide adenine dinucleotide phosphate (NADPH) is the product of this reaction. IDH1 has cytoplasmic localization, IDH3 as well as IDH2 exhibit a mitochondrial localization, and perform their function by favoring energy production in mitochondria [58]. Mutated IDH1 and IDH2 acquire neomorphic activity that converts α-KG to 2-HG, which has oncogenic properties and acts as a competitive inhibitor of enzymes involved in α-KG metabolism.

Finally, IDH exerts its influence on DNA methylation and histone modification. An example is the inhibitory effect on TET2 protein, which plays a role in epigenetic regulation [66]. This correlation has been well established in a study on AML patients, where global hypermethylation and loss of TET2 function has been observed [67].

### 2.2. Apoptosis Dysregulation Increases the Propensity to Malignant Transformation

Apoptosis—the programmed death of the cells—is a process characterized by a variety of cellular events, involving many components that are linked and dependent on each other at multiple levels, and able to directly or indirectly activate/inhibit certain molecular pathways [68]. Although both apoptotic (intrinsic and extrinsic) activation pathways have the same purpose and common endpoint (caspases), here we will only address the intrinsic pathway of apoptosis regulation because it directly involves the mitochondrial compartment. Given the fact that malignant neoplasms carry abnormal cell proliferation, detailed insight into the molecular map of apoptosis with certainty may answer many questions related to cancer etiology and therapy. Going backward in the scientific history, a milestone observation about the association between apoptosis and cancer formation is given by Weinberg et al. [69], when they unquestionably stated that one of the main characteristics of malignant transformation is the escaping from cell death—the hallmark of cancer, which further leads to uncontrolled cell proliferation.

By way of illustration, ineffective hematopoiesis induced by excessive apoptosis of hematopoietic precursors is one peculiar feature in MDS. During the early stage of MDS, apoptosis is increased through overexpression of TNFα, Fas-ligand, and TRAIL. Subsequently, the disease progresses because malignant cells acquire the ability to induce a shift in favor of anti-apoptotic and proliferative signals, escaping the pro-apoptotic pathways, with subsequent possible progression in acute myeloid AML [70,71].

In this context, the BCL-2 family proteins represent the key elements in the majority of the deregulated hematological apoptotic processes. These proteins were named after the discovery of the founding member B-cell lymphoma 2 BCL-2 protein, which is encoded by the *BCL2* gene. This gene was described for the first time as the second protein member in the B-cell lymphomas portrayed with the translocation t(14;18) [57,72,73]. Decades later, the scientific community, bearing in mind this assumption, developed a whole range of scientific investigations, having as their core observation the apoptotic dysregulation and malignant malformation. A number of factors, such as oxidative stress, DNA damage, high cytosolic calcium concentration, and oncogene activation, may initiate the mitochondrial pathway of apoptosis. The overall regulation of this pathway is mediated by the BCL-2 family proteins, which are divided into two major groups, pro-apoptotic (for example BAX, BAK, PUMA) and anti-apoptotic (for example BCL-2, BCL-XL, MCL-1) proteins [74,75].

Anti-apoptotic proteins like Bcl-2 and Bcl-xL were markedly upregulated in different types of leukemia malignancies, especially in more advanced MDS and in newly diagnosed AML, when compared to normal samples or AML patients under complete remission [76]. Their expression correlates with worse prognosis, while higher levels of proapoptotic proteins were associated with a lower risk of relapse or leukemic transformation [77]. Overexpression or mutations of BCL-2 allow chemoresistance to hematological malignancies. Nevertheless, other studies did not identify a significant association of BCL2 overexpression with prognosis, suggesting that Bcl-2 in AML blasts may not be a useful single factor affecting prognosis [76]. Interestingly, the high levels of BCL-2 identified in leukemic stem cells (LSCs) provide a novel strategy to target the quiescent LSC population. In this regard, in 2016, the specific BCL-2 inhibitor venetoclax (ABT-199), was approved by the Food and Drug Administration (FDA) for relapsed Chronic lymphocytic leukemia (CLL) with 17p deletion. Nowadays, venetoclaxis also used in AML patients. Another important player in leukemia apoptosis dysregulation, also crucial for hematopoietic development, is the anti-apoptotic MCL-1 protein. Its deregulated gene expression is associated with several hematological malignancies including AML, multiple myeloma (MM), and B-cell acute lymphoblastic leukemia (B-ALL). Gene deregulation is correlated with relapse and resistance to targeted therapies like venetoclax [78].

Integral portraiture of the connection between apoptosis and carcinogenesis is Cytochrome c and its role in caspases activation [79]. Pro-apoptotic factors increase the permeability of the OMM for cytochrome c, while the anti-apoptotic factors do the opposite [80]. Apoptosis begins with homo-oligomerization of BAK and BAX proteins, which create pores on the outer mitochondrial membrane in direct interaction with BH3-only proteins, the process thus termed ‘mitochondrial priming’ [81,82]. Throughout the process, the interaction of BH3-only proteins with pro-apoptotic factors enhances their oligomerization, thus achieving sufficient size and shape to form the effector pores on the outer mitochondrial membrane [83,84,85]. The anti-apoptotic factors manifest their activity by either binding to BH3-only protein and preventing its interaction with BAK and BAX or blocking already activated forms of BAX and BAK [81,86]. Venetoclax and several other promising BH3-mimetic drugs have been developed in order to inhibit the BH3-only proteins sequestration mediated by Bcl-2 and to promote apoptosis only via BAX-BAD-BH3 interaction. Within this scenario, Edlich et al. described how BAX localization determines the predisposition to apoptosis in human AML. Indeed, high BAX level in the blasts mitochondria correlates with increased sensitivity to chemotoxic stress, while high cytosolic BAX level is associated with cellular resistance to chemotherapies, suggesting BAX localization as a new important prognostic marker [87]. Reyna et al. develop BTSA1, a pharmacologically optimized BAX activator, which leads to sustained BAX-mediated apoptosis in leukemia cell lines and in AML xenografts, providing a novel therapeutic strategy in AML [88]. Thus, in conclusion, the formation of pores on the OMM and the penetration of cytochrome c can be named as a “point of the inevitable cell death”.

## 3. Mitochondria in the Hematopoietic Stem Cells: ADormant Galaxy

The two main features of stem cells are quiescence and lifelong self-renewal capacity [89]. The switch from this condition to cycling and proliferating is crucial for stem cell fate, homeostatic preservation, tissue regeneration, and it is directly linked to mitochondrial status [90,91]. The interplay between hematopoietic stem cells (HSC), microenvironment, and the hypoxic niche play a pivotal role in stem cell decision since it seemed to be enough for self-renewal or commitment decision [92]. Low oxygen levels sustained the characteristic metabolic state of HSC, and pointed out the adaptation to microenvironmental changes and plasticity property of stem cells [93]. The metabolic conditions of HSC are low energy, low ROS, and mitochondrial dynamics [94] (Figure 1). The balance between these features leads to the decision to remain quiescent or to differentiate, through the asymmetric or symmetric division process [95]. 

### 3.1. Low Energy Requirement Defines Hematopoietic Stem Cells

HSCs use anaerobic glycolysis as the main energy source while OXPHOS is used when HSCsswitch toward proliferation and differentiation [96]. Glycolysis is primarily used in order to adapt hypoxic conditions and to meet the cellular low energy demand [97]. During this process, glucose is converted into pyruvate and two ATP molecules are produced. The choice of glycolysis represents an HSC protective process against high levels of ROS production occurring during the OXPHOS process, under aerobic conditions, to produce energy via electrons transport. Specifically, glycolysis derived pyruvate enters into the mitochondrial tricarboxylic acid (TCA) cycle to produce 36 molecules of high-energy ATP [97]. From a biological point of view, a high level of hypoxia-inducible factor 1 (HIF-1) is a driving force for the glycolysis program activation [98]. HIF-1 is a transcription factor activated during hypoxia, after alpha subunit stabilization under low oxygen conditions. Subsequently, HIF1a interacts with a constitutively expressed HIF1b subunit to form a complex able to coordinate the transcription of genes involved in the glycolytic process [98]. Remarkably: (1) it promotes pyruvate dehydrogenase kinase (PDK2 and 4) expression. This event is followed by phosphorylation of pyruvate dehydrogenase (PDH) and suppression of pyruvate entrance in the TCA cycle [99]. In a knockout mice model of *HIF1a*, the overexpression on PDK2 and 4 restored the HSC number by increasing in glycolysis program and decreasing in mitochondrial metabolism, suggesting a pivotal role of HIF1-PDK axis for in vivo HSC maintenance [100]; (2) it induces the expression of genes coding for glycolytic enzymes, including Lactate Dehydrogenase A (LDHA) [101], Hexokinase 1 and 2 (HK1, HK2), Aldolase A and C (ALD-A, ALD-C), Enolase alpha (ENOalpha), and Phosphoglycerate Kinase 1 (PGK1) [102]; (3) it promotes the glucose transporters expression (GLUT1 and GLUT3) [103]. Upstream regulators of HIF1 are mainly expressed in hematopoietic stem cells and they are extremely relevant for HSC fate. MEIS1 and CBP/p300 interacting trans-activator 2 (CITED2) constitute the main transcriptional factor of the *HIF1* gene in the HSC compartment [104,105]. In turn, they directly control glycolysis and the oxidative stress response. Meis1^−/−^ mice showed a significant reduction in hypoxia-related genes, long-term HSCs exhaustion, and increasing in ROS levels [106].

Finally, fatty acid oxidation (FAO) is another alternative source of high-efficiency metabolic energy for HSC. This mechanism was activated in early hematopoietic cells and reduced during differentiation. FAO generates the production of NADPH and sustains ATP to counteract oxidative stress and to prevent ROS accumulation. A correct balance between pyruvate and beta-oxidation is crucial to maintain HSC property and to control the asymmetric cell division. Ito et al. identified a novel peroxisome proliferator-activated receptor δ (PPARδ)-FAO pathway as essential for the preservation of HSC [107]. PPARδ deletion led to a decrease in the amount of ATP in HSC, while treatment with etomoxir, a known inhibitor of mitochondrial β-oxidation, was associated with HSC exhaustion and inability to sustain the hematopoietic compartment. Moreover, PPAR-δ has shown to positively regulate FA oxidation by the PML-PPARδ-FAO axis [107]. Indeed, Pml^−/−^ defective HSCs are rescued through PPAR-δ activation, suggesting that dysfunctional PPAR-δ signaling and FAO are directly linked to PML. Finally, FAO-dependent NADPH production has been confirmed to have a pivotal role in the survival of leukemia cells, as described above.

### 3.2. Low Free Radicals Amount: A Peculiar Feature of Hematopoietic Stem Cells

It is well consolidated that OXPHOS activation is among those mainly responsible for ROS production and for the passing into the cell cycle [86,97]. According to the metabolic demands, the cell may decide to maintain stem features with low mitochondrial activity and low free radicals level, or to activate differentiation, with an increase in mitochondrial number and activity, through glycolysis or OXPHOS program, respectively [86,97]. Furthermore, ROS can damage many cellular elements, including RNA, DNA, and proteins. These events may be dangerous especially in HSCs, thus causing mutations harmful for maintenance of stem cell quiescence, finally leading to HSC senescence and loss of self-renewal capacity [108]. Therefore, HSCs activate several biological processes and redox sensors to reduce ROS damage and to avoid loss of self-renewal potential and HSC exhaustion as follows:

(1) Nuclear factor erythroid 2-Related Factor 2 (NRF2) preserves cellular redox balancing from oxidative stress. Its stability is tightly regulated in order to ensure a very short half-life of the protein [109]. Notably, under basal unstressed condition NRF2 is recruited byKelch-like ECH-associated protein 1 (KEAP1). The complex is retained into the cytoplasm and, after CUL3-E3 ubiquitin ligase activation, Nrf2 is rapidly ubiquitinated and degraded via proteasome [89]. In this scenario, one mechanism of cell response to toxic metabolites generated by ROS is the increase in NRF2 stability after the KEAP1 conformation change, leading to disaggregation of the NRF2-KEAP1 complex [110]. Subsequently, NRF2 translocates into the nucleus where it induces the transcription of several antioxidant enzymes, after the binding to the antioxidant response element (ARE), located on the promoter regions. Indeed, *Nrf2*^−/−^ reduced significantly the expression of superoxide dismutase genes (SODs) [111].

(2) Forkhead box O3 (FoxO3) are transcription factors crucial for HSCs maintenance [112,113]. They diminish ROS mutagenesis effects by regulating the expression of some genes encoding for detoxifying enzymes like peroxisome-located Catalase (CAT), superoxide dismutase 2 (SOD2), and the redox enzyme sestrin3 (SESN3) [114,115,116]. Moreover, it reduces mitochondrial activity by inactivation of mitochondrial respiratory chain proteins [114]. FoxO^−/−^ HSCs mice are defective in DNA damage repair and the expression of several anti-oxidant protein-encoding genes is significantly impaired, with a consequent increase in ROS levels [113]. The process described above is directly led by the upstream FoxO regulators PI3K-AKT and 5′ AMP-activated protein kinase(AMPK) signaling pathway, which acts as energetic and oxidative stress sensors. The AKT-mediated FOXO3a inactivation modulates ROS levels, the activation of the cycling process and the exhaustion of HSCs through a negative control exerted on Foxo3a and a positive effect on mTORC1 [117,118]. To counteract this effect, the Phosphatase and Tensin homolog (PTEN) antagonizes the PI3K/AKT pathway through an increase in phosphoinositide 4,5-biphosphate (PIP2) levels thereby protecting cells against oxidative damage [119]. In turn, increased ROS levels are directly linked to phosphoinositide-3,4,5-trisphosphate (PIP3) signaling as a result of oxidation of cysteine residues in PTEN catalytic site causing its inactivation [119]. Finally, *Pten*^−/−^ mice showed a depletion of the stem cell pool and increase in ROS level, suggesting an overlapping action with FoxO3a in HSCs maintenance [120]. In the second axis, AMP kinase (AMPK) positively activates FoxO3a by phosphorylation on different residues (Thr 179, Ser 399, Ser 413, Ser 555, Ser 588, and Ser 626), leading to the expression of genes involved in stress resistance [121]. AMPK, in turn, activates ATP generation through glycolysis and fatty acid oxidation and directly inhibits mTORC1 activity, reducing protein synthesis [122]. Importantly, FOXO3a activation mediated by AMPK exhibits a pro-autophagy gene expression pattern crucial to regulate dynamic mitochondrial numbers, with Unc-51 Like Autophagy Activating Kinase 1 (ULK1) as the main factor involved [123]. However, *AMPK*^−/−^HSC mice showed a modest dysfunction, suggesting a role of AMPK in HSCviability only under certain stress conditions.

Finally, FoxOs activity can be modulated by deacetylation on lysine residues conducted by Sirtuin1 (SIRT1) [124]. SIRTs are NAD+-dependent lysine deacetylases proteins, localized in mitochondria and activated by oxidative stress conditions. They are defined as metabolic sensors due to their ability to regulate cellular stress response. In the HSC compartment, SIRT1 activation induces ROS levels’ reduction and the increase in autophagy as well as in several anti-oxidant enzymes such as SOD and catalase. *SIRT1*^−/−^ HSC cells exhibit FoxO3a-mediated ROS level increase and quiescence reduction [125].

(3) Ataxia-telangiectasia mutated (ATM) kinase is a tumor suppressor involved in ROS regulation and in sustaining the hematopoietic stem cells pool [126]. It is a serine/threonine kinase, crucial for cellular response to double-strand DNA break damage activated by several mechanisms including accumulation of reactive metabolites [127]. *ATM*^−/−^mice showed increased ROS levels and functional reduction of HSCs [128]. Furthermore, the main consequence of *ATM* inactivation is the phosphorylation of p38 MAPK on the HSC compartment. This event induces the activation of p16, a cyclin-dependent kinase (CDK) inhibitor, and a subsequent reduction in HSCs self-renewal capacity and enhancement in cellular senescence [129]. Importantly, ATM is able to activate the BH3-Interacting Domain death agonist (BID), a BCL-2 family pro-apoptotic member [130]. Its phosphorylation by ATM kinase plays a pivotal role in maintaining the quiescence of HSCs and in protecting HSCs from oxidative stress. All these events suggest that ATM, by mediating ROS levels, may regulate cell fate decisions through p38 MAPK and BCL-2 proteins as downstream effectors. Finally, ATM activation by oxidative stress induces phosphorylation of NRF1, crucial for transcription of several antioxidant enzymes, and it mediates the phosphorylation of HIF1, with a consequent increase in HIF1 stability and activation of stem cells maintenance machinery [131,132].

### 3.3. Mitochondrial Dynamics Defines Hematopoietic Stem Cells

Mitochondria functionality is related to their morphology and the term “dynamics” is rightly used to describe the perpetual change in shape and number that in turn affects cellular reprogramming [133]. Therefore, mitochondrial high plasticity is necessary to improve cell viability, according to energy cellular demand and ATP levels, and they are a bona fide sensors of cell bio-energetic status [133]. Fusion, fission, and mitophagy are the main events involved in mitochondrial homeostasis. In stem cells niche ATP levels are low and mitochondria are in a basal activity state. They are localized in the perinuclear area, and their shape is generally fragmented, with fewer cristae but functional electron transport chain [134]. This conformation is strongly mito-fission dependent. The dynamin-related protein 1 (DRP1) and Mitochondrial fission 1 protein (FIS1) active form represent the main proteins responsible for this subcellular distribution pattern [134]. Impaired DRP1 and FIS1 expression induce a significant reduction in HSCs number due to a metabolic switch from glycolysis to OXPHOS that in turn promote a differentiating cell program [135]. It is fascinating that DRP1 and FIS1 seem to support the asymmetric stem cell division favoring the maintenance of stem cell feature only to daughter cells that receive more young mitochondria [136]. Lastly, a hypoxia condition activates the mito-fission pathway to maintain stem cell features while oxidative stress and ROS increased levels contribute to apoptosis or autophagy mito-fission dependent [137]. Along with the differentiation, cells are characterized by a mitochondrial-diffused distribution, with a tubular and elongated shape, in network with each other [138]. ATP levels produced via OXHPOS are high and ROS levels too. This feature is mito-fusion dependent, and Mitofusin 1/2 (MFN1, MFN2) and Mitochondrial Dynamin Like GTPase (OPA1) proteins are the main responsibility [139]. The MFN proteins are located in the OMMs while OPA1 in the IMMs. *Mfn1/2* depletion reduces the mitochondrial numbers and it induces HIF1 stabilization, thus facilitating glycolysis and pluripotency program [140]. In addition, mitochondrial fission and fusion factors reciprocally orchestrate efficient mitophagy. Mitophagy represents a crucial mechanism in stem cell maintenance [141]. It sustains functional mitochondrial, it promotes turnover of mitochondria, while removing those defective and it protects cells from apoptosis [141]. In addition, mitophagy is also important to regulate mitochondrial numbers according to cellular metabolic, thus it is essential for HSCs homeostasis. ROS high levels, by causing mitochondrial dysfunction, are the main train force to the mitophagy-clearance process [142]. Autophagy-related genes (*ATGs*) are responsible for mitophagy induction. *ATG* genes knockout inhibits mitophagy, induces aberrant mitochondrial accumulation, an increase of ROS levels and of the active metabolic state which impairs HSCs functionality [142].

Inhibition of mitochondrial fission through *DRP1* deficiency increases the clearance process leading to the loss in mitochondria numbers and culminates in cell death pathway after mitochondrial membrane potential (Δψm) depolarization [135]. Mito-fusion damage through *MFN 1* and *2* depletion provoke the accumulation of defective mitochondria [143]. The PTEN-induced putative kinase 1 (PINK1) and the Parkin-mediated pathway arbitrate directly the mechanism of mitophagy [144]. When the IMM becomes depolarized PINK1 is stabilized in the outer mitochondrial membrane surface, where it recruits the E3 ubiquitin ligase Parkin. Parkin ubiquitinates various substrates in the outer mitochondrial membrane, includingMFN1 and MFN2, favoring in turn mitophagy of damaged mitochondria [142,145].

The upstream tumor suppressors STK11/LKB1 (serine/threonine kinase 11) and their target, AMPK, contribute to HSCs quiescence by directly controlling mitochondrial autophagy, ULK1 and FIS1 mediated [146]. Inside this network, Foxo3a and Tuberous sclerosis 1/2 (TSC1/2), inhibition of the mTOR pathway, cooperate in the mitophagy process [123]. Lastly, PPARγ activation is crucial to preserve autophagy initiation, suggesting complex interconnection in HSC mitochondrial dynamics [147].

## 4. Leukemia Stem Cells Mitochondria: A Vulnerable Dormant Galaxy

Nowadays, leukemic stem cells (LSCs) represent the primary focus in the drug therapy field, since their targeting could lead to better outcomes as well as disease-free survival [148]. LSC cells are more resistant to first line therapies and persist over time, by raising the risk of relapse. The debate HCSs versus LSCs started in 1950 when Warburg and his group defined cancer stem cells (CSC) features, including anaerobic glycolysis, as the main force to produce ATP, and low-ROS conditions [149,150]. In the last few years, several studies confirmed that CSC uses preferentially OXPHOS [151,152] to produce energy, but maintains very low ROS levels through sustained mitophagy activation and significant high activity of the mitochondrial BCL-2 protein [153]. Moreover, it has been remarked that increased levels of OXPHOS in CSCs can support chemotherapy resistance [154]. Thus, understanding the unique and vulnerable properties of LSCs metabolic process could contribute to target leukemic stemness. Going back to the previous chapter, we highlighted that LSCs exhibit peculiar pathways to preserve their redox state, in order to sustain survival in hypoxic environments and to prevent them from going into apoptosis. Notably, LSCs display (1) OXPHOS, as a crucial process for cell survival and maintenance, despite the reduced cellular oxidative status and the inability to employ glycolysis (2) significant low ROS levels (3) aberrant BCL-2 expression and (4) elevated autophagy activation (Figure 1).

(1) The OXPHOS process is primarily used by LSCs and this feature could be exploited to target LSCs. Why LSCs favor OXPHOS instead of glycolysis is not well characterized, and this topic will represent an important avenue for future investigation. One possibility is that OXPHOS, being a highly efficient mechanism to produce energy, might be crucial for sustaining LSC energy requirements and survival [155]. Indeed, a common feature of LSCs is the ability to exploit other metabolic events such as protein/RNA synthesis, amino acid, and fatty acids metabolism as an essential source for energy production under metabolic stress conditions. Recent studies confirmed that amino acid and fatty acid metabolism (FAO) are altered in AML, heavily impaired LSCs survival when they are inhibited, and directly contribute to maintaining OXPHOS in LSCs [156,157].

Otherwise, leukemia stem cells are more addicted to these processes when compared to the counterpart HSC. Amino acids are commonly metabolized in the TCA cycle of LSCs derived from de novo AML patients [158]. Furthermore, targeting amino acid uptake results in decreased OXPHOS and selective targeting of LSCs. The fact that this pathway was highly sustained in leukemia stem cells, it represents a suitable druggablity target. Metabolism of specific amino acids including cysteine, glutamine, and branched-chain amino acids (BCAA) are the most important in the LSC compartment.

Glutamine is converted to glutamate through the enzyme glutaminase (GLS), which was found to be highly expressed in AML, and subsequently converted to α-KG. The α-KG crucial role in LSCs is linked to its ability to enhance the reduction of NADP+ to NADPH, which is required for the production of the antioxidant glutathione (GSH) and, in turn, protects LSC cells by neutralizing reactive oxygen species [159]. In vitro glutamine deprivation exhibit negative effects that can be rescued by adding α-KG to the AML cell, suggesting a prominent role of glutamine in LSCs survive [160]. Interestingly, inhibition of GLS synergized with venetoclax [160].

Cysteine drives energy metabolism in LSCs, since its metabolism mediates the synthesis of glutathione favoring oxidative phosphorylation and survival of LSCs. The depletion of cysteine targets directly OXPHOS in LSCs, confirming that LSCs in AML patients rely on amino acid metabolism to fuel oxidative respiration/phosphorylation [50]. Moreover, targeting amino acid uptake with Venetoclax (ABT-199), a promising BCL-2 inhibitor, decreased OXPHOS resulting in LSCs killing and achievement of remission in AML patients [161].

BCAAs are a group of three essential amino acids leucine, isoleucine, and valine, for which synthesis is catalyzed by the BCAA transaminases 1 (BCAT1) and 2 (BCAT2). BCAT1 overexpression seems to be a feature that is LSC specific, and it was associated with a worse prognosis [162]. In vitro BCAT1 inhibition significantly inhibits primary AML cell proliferation [163]. 

Similar to amino acid metabolism, FAO is a second important source of NADPH [164]. FAO directly generates a large number of fatty acyl–derived acetyl-coenzyme A (CoA) to promote the Krebs cycle and OXPHOS.FAO is involved in the mitochondrial pathways of the TCA [160] cycle and OXPHOS, where it contributes to the maintenance of self-renewal in LSCs [165,166]. Leukemia stem cells express a high level of fatty acid transporter CD36 [167]. The CD36+ LSCs exhibit a high level of FAO, quiescent feature, and chemotherapy resistance [167]. Treatment of LSCs and HSCs with a PPAR-δ inhibitor reduces stem cell properties HSCs/LSCs, confirming that the FAO process is crucial to maintain quiescence [168]. The carnitine palmitoyltransferase 1a (CPT1a) is a fatty acid transport that promotes the entry of fatty acid into the mitochondria matrix, crucial to carry out FAO. In this regard, a CPT1 inhibitor in combination with venetoclax and 5-azacytidine treatment, directly targets FAO in functional LSCs, suggesting that FAO at once contributes to LSCs [169]. Recent evidence suggests that FAO upregulation could contribute to therapy resistance in venetoclax-treated patients [170]. Thereby, FAO could contribute to a general mechanism in therapy resistance. Indeed, inhibition of fatty acid transport into the mitochondria leads to increased survival in AML mouse models [171]. Avocatin B, an avocado-derived compound, was demonstrated to possess potent anticancer activity by selectively targeting FAO, increasing ROS levels and chemosensitizing AML cells to Ara-C [172].

Finally, another interesting strategy used by LSC to maintain at the same time oxidative phosphorylation and functional mitochondria is the presence of gap junction (GJ) [173]. These structures have the ability to regulate leukemic stem/progenitor cell survival, proliferation and self-renewal by creating a physical interaction between LSCs and Bone Marrow stromal cells (BMSC). This type of communication creates continuous trafficking of whole functional mitochondria within the high oxidative stressed LSCs, which undergoes ROS detoxification and acquires the ability to escape from chemotherapy [173]. Connexins represent the bricks of these GJ and their deregulation has been associated with the reduction of functional HSCs. Connexin 43 (Cx43) is a crucial regulator of hematopoiesis and its expression increased in LSCs in order to directly regulate leukemic cells bioenergetics, energetic balance, and metabolism through functional mitochondria transfer [174]. Interestingly, leukemia cells generated in vitro without mitochondrial DNA (ρ0 cells) are characterized by a reduction of malignant potentiality, but this feature is completely modified and associated with leukemic progression when co-cultured with BMSC [175]. Inactivation of this process in a leukemic context could provide a novel approach in leukemia therapy.

(2) A broad number of recent studies confirmed that LSCs are characterized by a low cellular oxidative status, (ROS-low), due to enhanced activity in processes including autophagy and antioxidant machinery [176]. ROS-low cells possessed an engraftment advantage with shorter leukemia-free survival in transplanted mice. Sorting this ROS-low pool has allowed for observing some exclusive leukemia stem cell features [177]. LSCs are highly sensitive to increases in ROS levels and showed a reduced ability to use glycolysis to produce energy. AMPK is important for the survival of LSCs, where it controls mitochondrial dynamics and ROS level via autophagy ULK1-mediated, by conferring to LSCs the ability to sustain high mitochondrial stress [178]. AMPK deletion in AML LCS induced a significant ROS increase in the LSC compartment, followed by an increase in DNA damage and reduction in glucose flux [179]. Another relevant pathway important for ROS regulation in LSCs is the glutathione metabolism, a process hyper-activated in AML, whose expression correlates with an increase in relapsed and refractory AML. Glutathione peroxidase 3 (*GPX3*) ^−/−^is able to reduce LSCs via apoptosis, suggesting the importance of this process in LSC behavior [166]. Otherwise, increased expression of antioxidant enzymes promotes the progression from MDS toward secondary AML [180].

Lastly, HSCs and LSCs are highly sensitive to p53-mediated apoptosis, whose activation is heavily dependent on oxidative stress and ROS-high level. This implies that the downregulation of p53 response to basal ROS levels is crucial to the maintenance of hematopoietic stem cells pool [181]. Additionally, p53-induced gene 3 (PIG3) is a NADPH-dependent reductase, whose ability to induce ROS is essential for p53-mediated apoptosis in AML [182].

(3) LSCs have to maintain low ROS levels and high mitochondrial ATP levels, which are essential to sustain hematological disease. In addition, their mitochondrial membrane potential is reduced compared to HSCs [153,155]. Thus, the activation of anti-apoptotic responses is crucial to protect them from oxidative-stress conditions. In this scenario, hyper-activation of BCL-2 and increased autophagy mechanisms represent the main LSCs weapons, leukemia-specific, and therefore potentially druggable. BCL-2 inhibitors have proved to be efficient in exploiting ROS-low LSCs [153]. BCL-2 protein is able to block the apoptotic pathway by preventing oligomerization of BAX and BAK, and subsequently inducing mitochondrial outer membrane permeabilization. In addition, it is crucial for the OXPHOS process and for mitochondrial bioenergetics. After BCL-2 inhibitors treatment, ROS levels increased dramatically inducing apoptosis in almost exclusively LSC compartment [154,183]. Moreover, BCL-2 inhibitor drug Venetoclax associated with 5-azacitidine showed a broad spectrum of activity by inhibiting amino acid uptake and catabolism, by cooperating in autophagy processes impairment, thereby providing a molecular mechanism for targeting of LSCs [184].

Mitophagy is a mitochondrial quality control mechanism up-regulated in LSCs population, in order to efficiently eliminate damaged mitochondria, since oncogenes and high proliferation rate increase cellular stress. FIS1 comes out as crucial in the mitochondrial fission process and in autophagy activity to sustain LSCs [179]. In AML, LSC FIS1 expression is increased in comparison to the bulk of AML cells and to HSCs. Its expression correlates with an increase in AMPK activity, the direct regulator of FIS1. FIS1 inhibition, as other mitophagy actors’ inactivation, showed in AML cells a dramatic change in mitochondria morphology associated with a reduction in stem cell potential, proposing a critical role of mitophagy for sustaining LSCs [179]. Interestingly, higher expression of FIS1 is associated with increasing the risk of relapse and to chemo-resistance in AML. Notably, these patients exhibit a significant increase in the frequency of LSCs, thus confirming the pivotal role of autophagy in leukemia stem cell biology and drug resistance.

## 5. Mitochondria as Therapeutic “Hot-Spot” in Hematological Malignancies

By reviewing mechanisms of action of chemotherapy and molecular drugs that target mitochondria, we can indisputably say that most of them accomplish their action by acting on cell death pathways or cause cell death by disrupting metabolism (Figure 2). On the other hand, anti-apoptotic mechanisms are upregulated in the majority of hematological neoplasms, especially in those characterized by abnormal clonal proliferation. This new degree of cell proliferation certainly requires a redistribution of priorities in cellular metabolism. A milestone scientific discovery is presented in the study by Warburg et al. where the core discourse was about the increased metabolism of glucose to lactate in malignant cells to meet the needs for intensive cell growth [69,185]. More or less, the imbalanced cell growth mechanisms and genetic abnormalities highly present among hematological malignancies can be seen as potential foci for target treatment possibilities. Namely, AML or CLL may be observed as a role model to explain how mitochondrial target therapeutic action can be achieved. Here, we present the most important drug groups, already established or with high potential to be in use in the near future, through whose action, mitochondrial-targeted therapy can be described. Going into detail, the first barrier that preserves the integrity of mitochondria is the OMM, defined with selective permeability and absolute control of what exits and penetrates in and out of the mitochondrial matrix to the cytosol. The preservation of mitochondrial homeostasis is mainly due to the action of the protein permeability transition pore complex (PTPC). It is a massive protein structure composed of several parts such as VDAC located on OMM, adenine nucleotide translocase (ANT) located in IMM, and cyclophilin D (CYPD) in the mitochondrial matrix. Impairment of the function of any part of the PTPC will lead to the activation of intrinsic apoptotic pathways and certain cell death. The reported elements make the mitochondrial membrane barrier one of the first potential therapeutic “hot spots” [186].

A special group of substances called hexokinase (HK) inhibitors II repress the activity of the enzyme hexokinase, a catalytic enzyme that converts glucose to glucose 6 phosphate. This enzyme is closely correlated to the cytosolic part of the VDAC/ANT complex and, in addition to energy, together with the membrane protein complexes, it also has a regulatory function in terms of permeability [187]. It was already described that HK II is overexpressed in cancer cells. Because ANT activity is based on transferring the ATP molecule out of the mitochondria, blocking hexokinase leads to the cessation of glycolysis and to the absence of energy molecules transported into the cytosol, i.e., energy deficiency and cell death. A recent study [188] has explained the in vitro mechanism of two HK inhibitors II, arabinofuranosyl cytidine (Ara-C), and 2-Deoxy-D-glucose (2-DG). In fact, inhibition of glycolysis increases the sensitivity of certain AML cell lines (U937, OCI-AML3, THP-1, and KG-1) to Ara-C, and combinatorial therapy with 2-DG shows a synergistic effect. These positive results will further deepen scientific research in the field of HK inhibitors as potential pharmacological substances to target mitochondria. Speaking of AML and metabolic disruption, there are also scientific shreds of evidence about the role of mutated *IDH* in molecular pathogenesis [58,188,189]. The revolutionary discoveries and recent application of IDH1 and IDH2 inhibitors in the treatment of AML, although not directly targeted the mitochondrial structures, can still be considered in this light given the role of IDH isoforms in cellular metabolism [62,190].

Another way to alter mitochondrial metabolism is to inhibit energy production by OXPHOS or to induce hyperproduction of ROS. Both mechanisms of action will consequently lead to cell death. In order to avoid the volume of presentation, here we aimed to explain the basic functioning of this targeted therapy through some examples. Recent scientific research suggests that a complex I inhibitor—IACS-010759, has shown promising results in the treatment of patients with AML [191]. The realization that a certain type of micro-RNA is overexpressed and correlated with poor prognosis in some hematological neoplasms, has opened up opportunities to consider this option as a possible therapeutic approach [192]. By targeting these RNA molecules, it is disabled the translation into protein product and its implementation in the OXPHOS [193]. Going deeper into the layering of mitochondria, another protein complex in the IMM is the Bcl-2 family of proteins. Undoubtedly, it is the most exploited choice of mitochondrial target therapy. A large number of scientific studies are devoted to the Bcl-2 family of proteins and their role in cancer treatment. Bcl-2 inhibitors are one of the first in this group, especially used in CLL patients. By binding to the Bcl-2 protein, unpretentiously said, they block its anti-apoptotic activity or act on a gene/protein expression [194,195]. On the other hand, similar activity proclaims a group of drugs called BH3 mimetics [196,197]. They prevent the binding between BH3-only proteins and anti-apoptotic factors. Among these drugs, the most important one is Venetoclax and its application in CLL patients with 17p deletion, resulting in a loss of the *TP53* gene, enabling the direct apoptotic realization without the presence of p53 [81,184].

Furthermore, Venetoclax was also effective in relapsed/refractory AML patients [198,199]. Konopleva and colleagues report the important results obtained with a phase II trial of venetoclax monotherapy in a cohort consisting of older high-risk patients. In addition, the high levels of BCL-2 identified in LSCs provide a novel strategy to target the quiescent LSC population, crucial for driving the progression to AML. However, treatment with Venetoclax reported an increase of resistance mediated by the upregulation of alternative antiapoptotic proteins [200]. Within this scenario, dual inhibition seems a very attractive alternative in Venetoclax-resistant AML cells. Overexpression of mitochondrial protein MCL1, a common feature in hematologic malignancies, was associated with venetoclax-resistance in AML VU661013, a selective MCL1 inhibitor able to induce apoptosis in AML cells, overcoming venetoclax resistance [201]. The MAPK pathway directly stabilizes BCL-2 and contributed to developing resistance to Venetoclax [199]. Konopleva et al. confirmed that dual inhibition venetoclax-cobimetinib (GDC-0973), this latter an allosteric MEK inhibitor, reduced leukemia burden in vivo xenograft models. A synergistic effect, with astonishing results, was observed in phase I clinical trial with hypomethylating agents decitabine or 5-azacytidine [202]. This combination achieved an overall response rate of over 70% in a group of older high-risk patients AML. One potential target of hypomethylating agents could be p53 itself, or a member of the p53 family.

In this regard, we cannot continue the discussion on apoptosis-cancer therapy unless we take very careful consideration of the p53 properties—known in the scientific literature as the guardian of the genome [203,204]. It plays a major role in maintaining the normal dynamics of the cell cycle and DNA repair. Modern science has no doubt about the enormous impact of mutated p53 alleles on carcinogenesis. Literature data suggest that almost 50% of all cancers have a mutated p53 gene [205]. There are several ways in which p53 can be targeted as a potential anti-cancer treatment, both in general and in hematological diseases. Expression of p53 is dependent on MDM-2, a ubiquitin-protein ligase that stimulates the degradation of the wild-type and mutant p53 genes [206]. Considering the fact that there is an overexpression of MDM-2 in a large number of leukemias, lowering the level of MDM-2 will mean favoring cell death [207]. Thus, using targeting therapy that blocks the MDM2–p53 binding will thereby increase the p53 protein levels and promote cancer cell killing activity. Nutlin, for example, is a drug that exhibits this type of action [208]. The diverse role that p53 plays in apoptotic pathways, even though it has been a subject of study many decades, has not been adequately explained yet. p53 achieves its pro-apoptotic role by interacting with pro-apoptotic BH-3 only proteins, especially PUMA and NOXA. In addition, p53 in some cases may act as a BH-3 only protein and can directly activate the apoptosis effectors BAX and BAK [209].

Immunotherapy with p53 is probably the most promising target therapy that has received much attention in recent years in science. While this idea is not new, there are new ways of thinking about its implementation. At the core of its creation is the classic method of vaccine, i.e., immune stimulation with antibodies to the mutant p53 protein. This is quite possible since the mutated p53 protein can be perceived as an antigen by our immune system which would produce an appropriate antibody. Although the whole process is far from routine clinical application, it is interesting to mention some recent studies that presented promising scientific findings [210].

The most profound part of the mitochondria that can serve as a target “hot-spot” is the mitochondrial genome—mtDNA. Here we are talking about a genome that consists of 27 genes and the main product of its decoding is the proteins of the respiratory chain. Its activity through mtDNA is targeted by several pharmacological substances, and, in the therapeutic regimes of AML, the well-established is bleomycin. Bleomycin exerts its pharmacological action by causing mtDNA damage [211]. Enzymes that control DNA replication and translation can also serve as a therapeutic target. The very fact that cancer cells are characterized by abnormal proliferation, and therefore have intense mitochondrial liveliness, makes this therapeutic approach quite attractive and promising. 

## 6. Conclusions

Here, we painted a picture of the untangled mitochondrial galaxy. The intensive development of pharmacotherapy in the last few decades has opened new avenues in molecular pharmacology. Mitochondria are indispensable targets in this dynamic process, as one of the organelles where pharmacological action can be achieved. Given the role of mitochondria in apoptosis and apoptotic dysregulation in the course of carcinogenesis, the focus is particularly on their role as a “hot spot” in targeted cancer chemotherapy. The vital importance of mitochondria to cell life makes any segment of their activity a possible “dead point” that would endanger the cell’s fate, and the infamous “Perpetuum mobile” would be turned off. The key message emerging from this review is that leukemia cells, especially LSC, during disease progression gain the ability to achieve the largest amount of energy through increasing metabolic plasticity and through a significant microenvironment contribution. The OXHPOS, amino acid, and fatty acid metabolism help cancer cells to overcome the normal counterpart. Developing suitable strategies to eradicate LSCs by targeting their vulnerable and unique metabolic features can be exploited to improve novel therapeutic strategies. Within this scenario, Bcl-2 inhibitors represent the best novel approach to treat LSCs. However, more and more studies suggest that inhibition of a single mechanism will not be so efficient, and relapse is almost obvious. The encouraging results obtained withVenetoclax are most likely dependent on the fact that Bcl-2 seems to have multifaceted roles, including oxidative respiration, amino acid uptake, fatty acid oxidation, and apoptosis, all crucial for LSCs survival. Nevertheless, resistance mechanisms leading to the selection of novel resistant subclones can move researchers to identify novel combined therapy in order to explore and to target the dynamic leukemic metabolic interplay. Furthermore, peculiar LSCs mitochondrial characteristics remain to be elucidated, representing an exciting aim for future investigation.

## Figures and Tables

**Figure 1 ijms-21-03928-f001:**
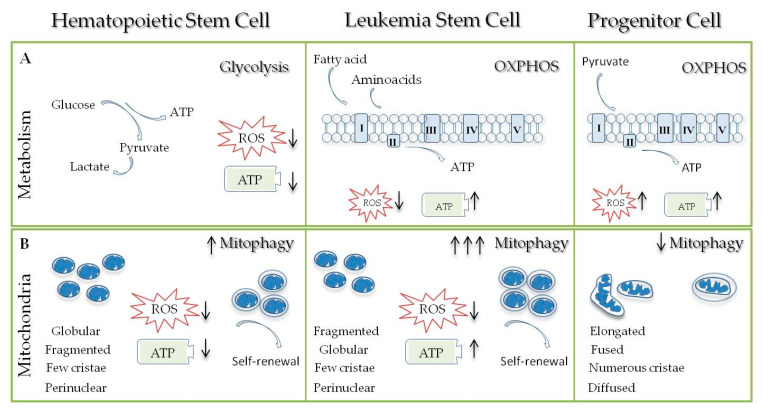
Metabolic regulation of Hematopoietic Stem Cells (HSCs), Leukemic Stem Cells (LSCs) and Progenitor Cells. (**A**) HSCs exhibit a low energy status, high glycolysis with resultant low ROS and low adenosine triphosphate (ATP) production. LSCs exhibit a low ROS level but high ATP production due to activation of oxidative phosphorylation (OxPHOS). Amino acid and fatty acid metabolism heavily contribute to maintain OXPHOS in LSCs. In contrast, progenitor cells exhibit high mitochondrial activity, high ROS, and ATP levels as a result of the OxPHOS process. (**B**) Hematopoietic stem cells have small motochondria, with globular and fragmented shape consistent with increased mitochondrial fission, with few cristae and localized in the perinuclear area. The mitophagy process represents a crucial mechanism in stem cell maintenance and self renewal. LSCs are characterized by high levels of autophagy in order to efficiently eliminate damaged mitochondria and reduce ROS levels since the rate of cellular stress is sustained. In contrast, progenitor cells have mitochondria with elongated and fused shape, consistent with increased mitochondrial fusion, and numerous cristae.

**Figure 2 ijms-21-03928-f002:**
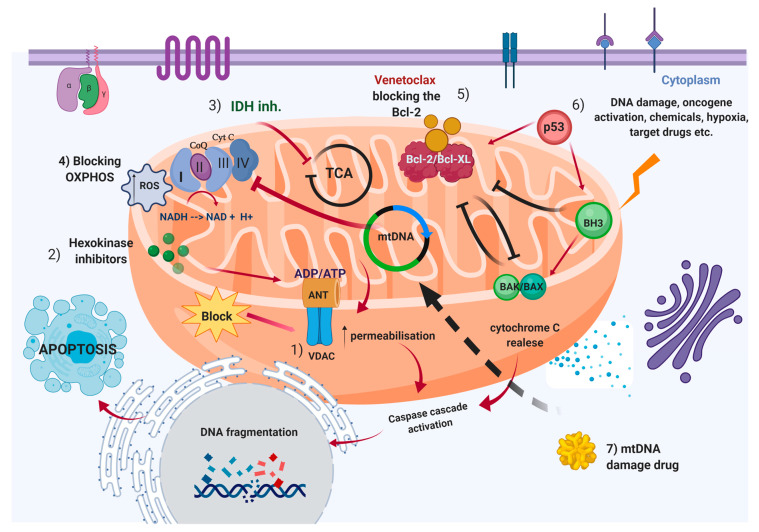
General overview of the mitochondrial targeted therapy approach. (**1**) Blocking VDAC selective permeability and activation of caspase cascade; (**2**) Hexokinase inhibitors block the enzyme hexokinase and ADP/ATP production; (**3**) IDH inhibitors manifest their activity inhibiting the α-ketoglutarate production in TCA; (**4**) Impaired OXPHOS and hyperproduction of ROS lead to cell death; (**5**) Using BCl-2 family proteins as targeted activation of intrinsic apoptotic pathways through the example of drug Venetoclax. Venetoclax releases the anti-apoptotic inhibition of Bcl-2 over BAX and BAK and then cytochrome c activiate the caspase cascade. In addition, as a response to extern stimuli or by drug, (**6**) p53 can be activated directly or in collaboration with the BH-3 only proteins can trigger the BAK/BAX pro-apoptotic activity. (7) mtDNA can be a core of action of drugs (such as bleomycin) which are producing damage in the DNA coding sequences and thereafter impaired cell function and death. (Picture created with Biorender.com).

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
