# Peer review of "Mitochondria: A Galaxy in the Hematopoietic and Leukemic Stem Cell Universe"

_ijms, 2020, doi:10.3390/ijms21113928_

Round 1

Reviewer 1 Report

This review is timely and comprehensive, bringing together multiple aspects of stem cell control that impact mitochondria and demonstrating that mitochondrial processes can be promising druggable targets for the selective removal of leukemia stem cells.

The style is occasionally unnecessarily flowery and the grammar inappropriate, so that there may be some issues for non-native speakers.

Section 1

The introduction presents mitochondria as dynamic organelles relevant to ROS, apoptosis and in close communication with ER. Their key function in energy production and as the site of the TCA cycle – the major intersection and exchange platform for a range of metabolic pathways should be made more clear.

L49 -. under “electrical gradient” most readers will probably understand the proton gradient that is established across the membrane as a result of electrons being transported along the electron transport chain. It may be that the authors are referring to the gradient of electronegativity along the ETC itself, that provides the driving force for the electron transfer between components. Given the potential for confusion here, it may be best to explain that the electron transport along the ETC drives the establishment of a proton gradient across the mitochondrial membrane and that the proton gradient then drives the phosphorylation of ADP to ATP.

Section 2

L93. It is too much to claim that mitochondrial dysfunction is the major underlying cause in all these diseases. I recommend that this be rephrased "…The list of diseases in which mitochondrial dysfunction represents the major underlying cause or makes a significant contribution is broad…"

L96 – the fact that interfering with mitochondria causes anemia is not for me a convincing indication that mitochondria have a pivotal role in leukemogenesis. In terms of energy and biosynthetic demands made on mitochondria, erythropoiesis is one of the most demanding processes in the body. It is not surprising that it is the first to be affected when mitochondria are dysfunctional. Erythropoiesis is similarly affected by vitamin deficiency.

L127 states that the impact of complex I mutations (decrease in NAD+ levels) may be similar to that of IDH mutations. Similarly, L133 repeats the claim that IDH1 and IDH2 mutations in AML manifest effects that are similar to complex 1 dysfunction. I do not see the connection here. The major impact of IDH1 or 2 in AML is to increase levels of 2 hydoxyglutarate (as explained by the authors), which then interferes with chromatin demethylation. This mechanism is explained in detail in the paper referenced (as ref 45), but is not mentioned by the authors. I do not see any support here for the implication that the leukemic effects of IDH1/2 mutations have to do with NAD+ Levels*.

*A potential explanation is offered later (L177-178) where it is stated that high levels of 2 hydroxyglutamate (product of mutated IDH1/2) reduce the activity of complex IV. I can see that one consequence of this would be a back-up in electron transport and a reduction in the NAD+/NADH ratio (in common with complex I mutations). However, whether or not 2hydroxyglutarte accumulation in IDH1/2 mutated cells results in NAD+ accumulation, the more established mechanism of leukemogenesis driven by increased chromatin methylation (and inactivation of tumor-suppressor genes) should at least be mentioned.

L245 The explanation of mtiochondrially-triggered apoptosis is good, but it is too much to claim that the balance between pro and anti-apoptotic factors is the “supreme juncture from which carcinogenesis begins”. Carcinogenesis can begin with any one of a number of lesions. Changes in the regulation of apoptosis are often a downstream event.

Section 3

L262 and L266 repeated sentence “The choice of glycolysis represents a HSC protective….”

Section 4

Figure 1

The notation used for ATP generation is not consistent. In Fig 1A (Hematopoietic stem cell) the net gain of 2ATP equivalents through glycolysis is shown as the sum of the synthesis (phosphorylation) steps: 2ADPs are phosphorylated to 2ATP. The “investment” form of this equation would read 2ATP (invested) generate 4 ATP (payout)

For the Leukemic Stem Cell and Progenitor Cell panels the equation shown (2ADP generates 36 ATP) is neither the sum of the synthesis steps nor the investment form.

Given the potential for confusion, I suggest deleting the input ATP/ADP, simply leaving the arrows and the product ATPs.

Author Response

Response to Reviewer 1 Comments

Dear Editor,

We truly thank the referees for their constructive comments, useful for the manuscript improvement. We hope that having addressed their comments we have strengthened our paper. Please note that unless otherwise stated, all page and figure numbers refer to those of the revised manuscript. The reviewer’s comments are copied in italics.

In this resubmission, we have summarized below how we have addressed all the comments
raised by the reviewers.

Response to Reviewer 1

We appreciate the positive and valuable feedback from the reviewer.

  1. The introduction presents mitochondria as dynamic organelles relevant to ROS, apoptosis, and in close communication with the ER. Their key function in energy production and as the site of the TCA cycle – the major intersection and exchange platform for a range of metabolic pathways should be made more clear. 

As suggested by the reviewer, we supplemented the introduction by explaining the function of mitochondria in terms of the TCA cycle (L85-93, page: 2). We have avoided a more extensive discussion on this topic to emphasize the function of mitochondria in hematopoietic and leukemic stem cells, which is the main goal of our review.

  1. L49 -. under “electrical gradient” most readers will probably understand the proton gradient that is established across the membrane as a result of electrons being transported along the electron transport chain. It may be that the authors are referring to the gradient of electronegativity along with the ETC itself, that provides the driving force for the electron transfer between components. Given the potential for confusion here, it may be best to explain that the electron transport along the ETC drives the establishment of a proton gradient across the mitochondrial membrane and that the proton gradient then drives the phosphorylation of ADP to ATP. 

According to the reviewer advice in the present revised version, we have made more clear the process of electron transfer along with the inner mitochondrial membrane (L83-85, page: 2)

  1. It is too much to claim that mitochondrial dysfunction is the major underlying cause of all these diseases. I recommend that this be rephrased "…The list of diseases in which mitochondrial dysfunction represents the major underlying cause or makes a significant contribution is broad…" 

As suggested by the reviewer in the present revised version we have modified the sentence (rephrased it) to be more clear and accurate (L144, page: 3) 

  1. L96 – the fact that interfering with mitochondria causes anemia is not for me a convincing indication that mitochondria have a pivotal role in leukemogenesis. In terms of energy and biosynthetic demands made on mitochondria, erythropoiesis is one of the most demanding processes in the body. It is not surprising that it is the first to be affected when mitochondria are dysfunctional. Erythropoiesis is similarly affected by vitamin deficiency. 

We frankly thank the reviewer for this comment. In the current version, we have modified and implemented this section according to the reviewer's advice by adding some more precise explanations and some new references. (L152-162, pages: 3-4) 

  1. L127 states that the impact of complex I mutations (decrease in NAD+ levels) may be similar to that of IDH mutations. Similarly, L133 repeats the claim that IDH1 and IDH2 mutations in AML manifest effects that are similar to complex 1 dysfunction. I do not see the connection here. The major impact of IDH1 or 2 in AML is to increase levels of 2 hydoxyglutarate (as explained by the authors), which then interferes with chromatin demethylation. This mechanism is explained in detail in the paper referenced (as ref 45), but is not mentioned by the authors. I do not see any support here for the implication that the leukemic effects of IDH1/2 mutations have to do with NAD+ Levels*.

*A potential explanation is offered later (L177-178) where it is stated that high levels of 2 hydroxyglutamate (product of mutated IDH1/2) reduce the activity of complex IV. I can see that one consequence of this would be a back-up in electron transport and a reduction in the NAD+/NADH ratio (in common with complex I mutations). However, whether or not 2hydroxyglutarte accumulation in IDH1/2 mutated cells results in NAD+ accumulation, the more established mechanism of leukemogenesis driven by increased chromatin methylation (and inactivation of tumor-suppressor genes) should at least be mentioned. 

We genuinely and truly thank the referee for this articulated and helpful comment. To properly address this issue the herein revised version contains certain changes related to the role of IDH in the carcinogenesis of hematological malignancies. We sincerely believe that in this way we have managed to meet the reviewer request (L247-264, page: 5)  

  1. L245 The explanation of mtiochondrially-triggered apoptosis is good, but it is too much to claim that the balance between pro and anti-apoptotic factors is the “supreme juncture from which carcinogenesis begins”. Carcinogenesis can begin with any one of a number of lesions. Changes in the regulation of apoptosis are often a downstream event.

According to reviewer suggestion we deleted the last sentence of section 2.2 (L364, page: 6) and we modified the section 2.2 title (L265, page: 5).

  1. L262 and L266 repeated sentence ”The coiche of glycolysis represents a HSC protective…”

We truly thank the referee and we apologize for the mistake. We corrected by deleting the second sentence (L384, page: 7).

  1. The notation used for ATP generation is not consistent. In Fig 1A (Hematopoietic stem cell) the net gain of 2ATP equivalents through glycolysis is shown as the sum of the synthesis (phosphorylation) steps: 2ADPs are phosphorylated to 2ATP. The “investment” form of this equation would read 2ATP (invested) generate 4 ATP (payout). For the Leukemic Stem Cell and Progenitor Cell panels the equation shown (2ADP generates 36 ATP) is neither the sum of the synthesis steps nor the investment form. Given the potential for confusion, I suggest deleting the input ATP/ADP, simply leaving the arrows and the product ATPs.

In the current version the Figure 1 has been modified according to the reviewer suggestion.

Reviewer 2 Report

The current manuscript by C.P. et al focuses on the role of mitochondrion in HSCs and LSCs. The authors begin with a brief introduction of mitochondrion and its function followed by its role in disease development. The authors then give a detailed summary of mitochondria's roles in HSC maintenance, leukemia development and the drug resistance of LSCs. The authors conclude the manuscript with a focus on targeting mitochondrion in AMLs. The authors have cited lots of publications and the manuscript is well-written. However, there are several points that the authors may want to address:

Major points:

  1. There are several good reviews on the role of mitochondrion in disease development including cancer. It does not seem that the current version of manuscript is a standout. Instead of focusing on the generally well-known regulators of Oxphos, mitofission/fusion and mitophagy, the authors may consider introduce more on the biochemical properties of mitochondrion such as mitochondrial metabolism of lipids (cardiolipin, sphinogolipids, phospholipids et al) and amino acids production by ETC and their role in normal (HSCs) or disease (AML) settings. 
  2. In Section 2.2, the evidence provided for carcinogenesis is too less. Either changing the section title or adding more evidence is advised.
  3. In the 3rd section, the authors mention that HSCs mainly depends on glycolysis for ATP production and HSCs possess more fission mitochondria and a low ROS state. However, HSCs are also efficiently using fatty acids by fatty acid oxidation in mitochondrion. Thus, the author might want to explain the contradiction.
  4. Cancer cells can survive without the normal functions of mitochondrion (ρ0 cells). However, when engrafted, these cells will acquire mitochondrial DNA or even mitochondrion from niches. The authors might want to add this point.
  5. The term 'overall low metabolic rate' in LSCs is incorrect. LSCs might have lower ATP levels, but there are other metabolic events heavily going on such as protein/RNA synthesis, fatty acids metabolism et al.

Minor points:

  1. Page 3, 'Functional abnormalities in complex I... capacity.' Complex I abnormalities can lead to high ROS, but could it lead to low ROS (such as loss of function)? 
  2. Page 3, 'the impact of these mutations, being...IDH mutations.' How is decrease in NAD+ similar to IDH mutations? IDH mutations actually produced NAD(P)+.
  3. Page 10, 'additionally, p53...' this sentence does not read correctly both grammarly and scientifically. And also please cite related publication for this conclusion.
  4. References 140 and 166 are in wrong places.

Author Response

Response to Reviewer 2 Comments

Dear Editor,

We truly thank the referees for their constructive comments, useful for the manuscript improvement. We hope that having addressed their comments we have strengthened our paper. Please note that unless otherwise stated, all page and figure numbers refer to those of the revised manuscript. The reviewer’s comments are copied in italics.

In this resubmission, we have summarized below how we have addressed all the comments
raised by the reviewers.

Response to Reviewer 2

We appreciate the positive feedback from the reviewer.

  1. There are several good reviews on the role of mitochondrion in disease development including cancer. It does not seem that the current version of manuscript is a standout. Instead of focusing on the generally well-known regulators of Oxphos, mitofission/fusion and mitophagy, the authors may consider introduce more on the biochemical properties of mitochondrion such as mitochondrial metabolism of lipids (cardiolipin, sphinogolipids, phospholipids et al) and amino acids production by ETC and their role in normal (HSCs) or disease (AML) settings. 

We frankly thank the reviewer for the comment. In the current version we have implemented, articulated and detailed the introduction by adding a new section (L65-137 page: 2), according  to the reviewer advices.

  1. In Section 2.2, the evidence provided for carcinogenesis is too less. Either changing the section title or adding more evidence is advised.

According to reviewer suggestion, we modified the section title (L265, page: 5) and deleted the last sentence of this section (L364, page: 6).

  1. In the 3rd section, the authors mention that HSCs mainly depends on glycolysis for ATP production and HSCs possess more fission mitochondria and a low ROS state. However, HSCs are also efficiently using fatty acids by fatty acid oxidation in mitochondrion. Thus, the author might want to explain the contradiction.

We appreciate the referee for this articulated and helpful comment. In the first version of the manuscript we decide to strengthen the importance of fatty acid oxidation in LSC metabolism, even if we are well aware of their importance in normal HSC. Given the potential for confusion and to properly address this issue we implemented the section with the importance of fatty acid oxidation in mitochondrion of HSC (L403-434, pages: 7-8).

  1. Cancer cells can survive without the normal functions of mitochondrion (ρ0 cells). However, when engrafted, these cells will acquire mitochondrial DNA or even mitochondrion from niches. The authors might want to add this point.

To fulfill the reviewer request the revised version of the manuscript has been implemented with a comment about this fascinating mechanism, really useful to leukemia cells drug resistance. (L635-659, pages: 11-12)

  1. The term 'overall low metabolic rate' in LSCs is incorrect. LSCs might have lower ATP levels, but there are other metabolic events heavily going on such as protein/RNA synthesis, fatty acids metabolism et al.

We truly thank the referee and we apologize for the mistake. We modified this section by changing the sentence (L577-579, page: 10). In addition, we further strengthened the importance of other metabolic events to sustain LSC energy needs, by implemented the subsequent section (L580-662, pages: 10-12).

Minor point:

  1. Page 3, 'Functional abnormalities in complex I... capacity.' Complex I abnormalities can lead to high ROS, but could it lead to low ROS (such as loss of function)? 

According to reviewer suggestion we add a new sentence (L180-184, page: 4)

  1. Page 3, 'the impact of these mutations, being...IDH mutations.' How is decrease in NAD+ similar to IDH mutations? IDH mutations actually produced NAD(P)+

We thank the referee for this comment. The revised version contains certain changes related to the role of IDH in the carcinogenesis of hematological malignancies and its relation with electron transport chain complex. We sincerely hope that in this way we have managed to meet the reviewer request (L247-264, page: 5)  

  1. Page 10, 'additionally, p53...' this sentence does not read correctly both grammarly and scientifically. And also please cite related publication for this conclusion.

According to the reviewer suggestion we modified this sentence (L675-679, p: 12). We sincerely believe that in this way we have managed to meet the reviewer request.

  1. References 140 and 166 are in wrong places.

We apologize for the mistake and we modified these references according to reviewer suggestion.

Round 2

Reviewer 2 Report

The authors did a beautiful job on revising of the manuscript. Therefore, I think it is suitable for publication in IJMS.